# Assessing the Feasibility of Using Electrochemical Skin Conductance as a Substitute for the Quantitative Sudomotor Axon Reflex Test in the Composite Autonomic Scoring Scale and Its Correlation with Composite Autonomic Symptom Scale 31 in Parkinson’s Disease

**DOI:** 10.3390/jcm12041517

**Published:** 2023-02-14

**Authors:** Yu-Chuan Huang, Chih-Cheng Huang, Yun-Ru Lai, Chia-Yi Lien, Ben-Chung Cheng, Chia-Te Kung, Yi-Fang Chiang, Cheng-Hsien Lu

**Affiliations:** 1Departments of Neurology, Kaohsiung Chang Gung Memorial Hospital, Chang Gung University College of Medicine, Kaohsiung 83301, Taiwan; 2Departments of Hyperbaric Oxygen Therapy Center, Kaohsiung Chang Gung Memorial Hospital, Chang Gung University College of Medicine, Kaohsiung 83301, Taiwan; 3Departments of Internal Medicine, Kaohsiung Chang Gung Memorial Hospital, Chang Gung University College of Medicine, Kaohsiung 83301, Taiwan; 4Departments of Emergency Medicine, Kaohsiung Chang Gung Memorial Hospital, Chang Gung University College of Medicine, Kaohsiung 83301, Taiwan; 5Department of Biological Science, National Sun Yat-sen University, Kaohsiung 80424, Taiwan; 6Department of Neurology, Xiamen Chang Gung Memorial Hospital, Xiamen 361126, China

**Keywords:** modified Composite Autonomic Scoring Scale, COMPASS 31, Sudoscan, Parkinson’s disease

## Abstract

The Composite Autonomic Scoring Scale (CASS) is a quantitative scoring system that integrates the sudomotor, the cardiovagal, and the adrenergic subscores, and the Composite Autonomic Symptom Scale 31 (COMPASS 31) is based on a well-established comprehensive questionnaire designed to assess the autonomic symptoms across multiple domains. We tested the hypothesis that electrochemical skin conductance (Sudoscan) can be a substitute for the quantitative sudomotor axon reflex test (QSART) in the sudomotor domain and assessed its correlation with COMPASS 31 in patients with Parkinson’s disease (PD). Fifty-five patients with PD underwent clinical assessment and cardiovascular autonomic function tests and completed the COMPASS 31 questionnaire. We compared the modified CASS (integrating the Sudoscan-based sudomotor, adrenergic, and cardiovagal subscores) and CASS subscores (the sum of the adrenergic and cardiovagal subscores). The total weighted score of COMPASS 31 was significantly correlated with both the modified CASS and the CASS subscore (*p* = 0.007 and *p* = 0.019). The correlation of the total weighted score of COMPASS 31 increased from 0.316 (CASS subscores) to 0.361 (modified CASS). When we added the Sudoscan-based sudomotor subscore, the case numbers for autonomic neuropathy (AN) increased from 22 (40%, CASS subscores) to 40 (72.7%, modified CASS). The modified CASS not only better reflects the exact autonomic function, but also improves the characterization and quantification of AN in patients with PD. In areas in which a QSART facility is not easily available, Sudoscan could be a time-saving substitution.

## 1. Introduction

The Composite Autonomic Scoring Scale (CASS) is a widely used quantitation score for grading the autonomic impairment that integrates the sudomotor, the cardiovagal, and the adrenergic subscores [1]. The result of the quantitative sudomotor axon reflex test (QSART) is used to code the subscore in the sudomotor domain. However, QSART facilities are not commonly available. In some countries such as Taiwan, poor income is a considerable obstacle to the more widespread clinical use of this test. In laboratories where QSART is unavailable, the focus of the examination is limited to cardiovascular autonomic function. In such circumstances, only part of the patient’s CASS, the subscores in the cardiovagal and adrenergic domains, can be obtained, and that of the sudomotor domain is lacking.

The Sudoscan is a new device developed to provide a simple, rapid, and quantitative assessment of sudomotor function. The electrochemical skin conductance (ESC) measured by the Sudoscan has been proven to correlate with the skin nerve fiber density [2]. The use of the Sudoscan in screening for diabetic neuropathy has been validated [3,4].

In this study, we tried to develop a sudomotor score based on ESC measured using the Sudoscan. Because the availability of the Sudoscan is greater than that of the QSART, for laboratories where QSART is unavailable, the newly developed score may be a suitable substitute for the sudomotor subscore of the CASS. Specifically, by using the Sudoscan-based score, we can obtain a modified CASS with which the autonomic functions, including the sudomotor, the cardiovagal, and the adrenergic domains, are all covered.

The Composite Autonomic Symptom Score (COMPASS), which was developed at the Mayo Clinic, is a validated 84-question scoring instrument for the assessment of autonomic symptom severity [5]. The use of the COMPASS is time consuming because of the significant number of questions in the instrument. Therefore, a refined and abbreviated version, the COMPASS 31, which consists of 31 questions in 6 domains (orthostatic intolerance, vasomotor, secretomotor, gastrointestinal symptom, bladder, and pupillomotor), was developed [6]. The COMPASS/COMPASS 31 and the CASS have been demonstrated to be useful instruments for the differential diagnosis and follow-up for patients with parkinsonism [5,7,8].

Autonomic impairment is an important non-motor feature in patients with Parkinson’s disease (PD) [9]. According to our hypotheses, the modified CASS would better reflect the “whole picture” of autonomic function compared to only part of the subscores (the sum of the cardiovagal and the adrenergic scores (the CASS subscores)), in patients with PD, and would also improve the characterization and quantification of autonomic neuropathy (AN) in patients with PD.

## 2. Materials and Methods

### 2.1. Study Design and Patient Selection

Patients with a definitive diagnosis of idiopathic PD according to clinical diagnostic criteria, as well as those who had undergone magnetic resonance imaging for the diagnosis of PD [10,11], were recruited. The exclusion criteria were: (1) advanced PD stage (Hoehn and Yahr staging equal to or more than 4, and an inability to walk independently); (2) severe cognitive impairment leading to an inability to follow instructions; (3) having any known cardiovascular events such as stroke or myocardial infarction; (4) having diabetes; and (5) having pacemaker implantation or any type of arrhythmia that would make the assessment of cardiovagal function infeasible. The study was approved by the hospital’s Institutional Review Committees on Human Research (IRB 201901802B0). All participating patients received verbal and written information about the purpose of this research, and subsequently signed their informed consent.

### 2.2. Clinical Assessment of PD

All patients gave their complete history whereby we recorded their age at disease onset; their sex; their body height and body weight; the disease duration; and medication information including the daily levodopa dosage (expressed as levodopa equivalent dose, LED) [12]. The clinical assessments, including the Hoehn and Yahr stage [13] and Unified Parkinson’s Disease Rating Scale (UPDRS) [14], were performed during the “off” state, which was defined as 12 h after the latest dose of the anti-Parkinsonism agents [15]. Finally, the COMPASS 31 questionnaire [6] was used to assess the patients’ autonomic symptom profiles.

### 2.3. Testing Autonomic Function

All of the patients with PD underwent a standardized evaluation of cardiovascular autonomic function, as described by Low [16]. The testing was performed in the “off” state. The tests included detecting the heart rate response to deep breathing (HRDB), the Valsalva maneuver (VM), and the 5-min head-up tilt test. After the cardiovascular autonomic testing, the sudomotor function was then assessed using the Sudoscan following a previously described method [4]. No coffee, food, alcohol, or nicotine was permitted within the 4 h before the tests. Patients on medications known to cause orthostatic hypotension or to otherwise affect autonomic testing results were asked to stop the drug for five half-lives if it was not harmful to the patient’s well-being.

Heart rate (HR) was derived from continuously recorded standard three-lead ECG (Ivy Biomedical, model 3000; Branford, CT) while continuous blood pressure (BP) was recorded using beat-to-beat photoplethysmographic recordings (Finameter Pro, Ohmeda; Englewood, OH, USA). Subsequently, the parameters of the HRDB and the Valsalva ratio (VR) were obtained through the computation conducted by Testworks (WR Medical Electronics Company, Stillwater, MN). The detailed computation is described by Low [16]. In addition, baroreflex sensitivity (BRS) was derived from changes in the HR and the blood pressure during the early phase II of VM by applying the least-squares regression analysis (BRS_VM). As for the sympathetic sudomotor function, the ESCs of the hands and feet were obtained using the Sudoscan. The scoring system for the Sudoscan was modified from the original CASS scores for sudomotor domains [1]. AN was defined as a minimum score of 1 in at least two of the CASS domains (cardiovagal, sudomotor, or adrenergic) or a minimum score of 2 in one domain according to Low’s study [17]

### 2.4. Grading the Severity of Autonomic Impairment

The severity of a patient’s cardiovascular autonomic impairment was graded using the cardiovagal and adrenergic subscores of the Composite Autonomic Scoring Scale (CASS) [1]. Because the QSART was not used, the sudomotor subscore was unavailable. Instead, a Sudoscan-based sudomotor subscore, in which 3 points could be given, was created, as described in Table 1. According to the method by Dyck et al. [18], the 97.5 percentile of the normative data of our own laboratory was set as the cut-off value of “abnormal” for the Sudoscan measurement. The values were 51.5 and 60.1 microsiemens (µS) for the hand and the foot, respectively (unpublished data). Combined with the cardiovagal and the adrenergic subscores, a modified CASS, which could give 10 points analogous to the original, was obtained.

### 2.5. Statistical Analysis

Data are expressed as mean ± SD or median (interquartile range (IQR)) for continuous variables and as median [minimum, maximum] for ordinal variables. Associations between measurements were evaluated using Pearson correlation tests for normally distributed continuous data or using the Spearman non-parametric test for continuous data with skewness or for ordinal data. Furthermore, receiver operating characteristic (ROC) curves were generated to determine the significance of the COMPASS 31 score and predict the presence of AN. The statistical significance threshold was set at 0.05. All statistical analyses were conducted using IBM SPSS Statistics v23 statistical software (IBM, Redmond, WA, USA).

## 3. Results

### 3.1. General Characteristics of Patients with PD

Table 2 shows the general characteristics of our enrolled patients, including their age, their sex, their BMI, their waist circumference, the disease duration, the UPDRS scores, the COMPASS 31 scores, and information about medication.

### 3.2. Correlations between COMPASS 31 and Different CASS Scoring Systems

The results of the correlation analyses between the COMPASS 31 and the different CASS scoring systems are listed in Table 3. The total weighted score of the COMPASS 31 is significantly correlated with both the modified CASS and the CASS subscores (*p* = 0.007 and *p* = 0.019, respectively) (Figure 1). The correlation of the total weighted score of the COMPASS 31 increased from 0.316 (CASS subscores) to 0.367 (modified CASS).

### 3.3. Comparison of Functional Scores, COMPASS 31 Scores, and Autonomic Function between Patients with and without AN Based on Different CASS Scoring Systems

When we added the Sudoscan-based sudomotor subscore, the case numbers for autonomic neuropathy (AN) increased from 22 (40%, CASS subscores) to 40 (72.7%, modified CASS). The clinical scores, the COMPASS 31 scores, and the cardiovascular autonomic function between patients with and without modified CASS-based AN are listed in Table 4. As for the COMPASS 31 scores, the orthostatic intolerance score and the total weighted COMPASS score were significantly different between the two groups (*p* = 0.031 and *p* = 0.003, respectively). The differences between HRDB, the ESC values for hands and feet, and the ESC cardiovascular autonomic neuropathy (CAN) risk scores also showed statistical significance between the groups (*p* = 0.022, *p* = 0.002, *p* < 0.0001, and *p* = 0.028, respectively). The values of the VR and the BRS_VM were larger in the group without AN than with AN, but the difference was not statistically significant.

### 3.4. ROC Analysis for COMPASS 31 Score in Predicting Modified CASS-Based AN

The diagnostic accuracy for the COMPASS 31 score in the modified CASS-based AN was analyzed using an ROC curve. The area under the ROC curve for the COMPASS 31 score was 0.715 (*p* = 0.02, 95% CI = 0.55–0.88). The cut-off value of AN diagnostic accuracy for the COMPASS 31 score was 12.45 (sensitivity = 74% and specificity = 67%) (Table 5 and Figure 2).

## 4. Discussion

CASS is a widely used quantitation score for grading autonomic impairment with subscores for sudomotor, cardiovagal, and adrenergic deficits [1]. The American Academy of Neurology’s evidence-based guidelines for clinicians recommend that a combination of autonomic screening tests in the CASS should be considered to achieve the highest diagnostic accuracy for AN [19]. Nevertheless, in some countries such as Taiwan, few autonomic laboratories have access to the QSART. As previously mentioned, poor income is a considerable obstacle to the more widespread clinical use of this test. In addition, it is difficult to maintain the equipment in Taiwan because of the humid weather. The desiccant must be changed frequently or the sensor will be out of order due to the high humidity. Therefore, for laboratories in which the QSART is unavailable, only the subscores in the cardiovagal and adrenergic domains can be used. However, the autonomic impairment in each domain could be different. For example, some diseases/disorders may affect only certain parts of the autonomic system [20]. Using only the cardiovascular autonomic function to present the severity of the “whole” autonomic function may be overgeneralization. By using the modified CASS, the sudomotor domain can be covered in addition to cardiovascular autonomic function. Moreover, the scoring system allotting 10 points is likely to have better discrimination than that with only 7 points. In this study, we provided a Sudoscan-based sudomotor subscore to replace the QSART-based sudomotor subscore for laboratories without the QSART facilities, because the former is more accessible than the latter.

The QSART is a well-established test for the evaluation of sympathetic sudomotor function [21]. Nevertheless, the Sudoscan is more commonly available than the QASRT due to its lower cost and easier maintenance. Both the QSART and the Sudoscan use sweat glands as a proxy to evaluate sudomotor function. There are two main differences between these tests: they use different stimulant methods and different measurements. The QSART uses a chemical stimulus (acetylcholine) while the Sudoscan uses an electrical one, and the QSART measures the sweat volume while the Sudoscan measures chloride ion flow in the skin. The results of the two testing methods are not significantly correlated [2]. The disruption of the correlation between these two measurements could be attributed to the variation in sodium and chloride concentrations due to individual characteristics, and environmental and dietary factors [22]. A study by Novak [2] showed a significant correlation between the ESC measured using the Sudoscan and skin biopsy with quantitation of the intraepidermal nerve fiber density, which is still recognized as the gold standard. Furthermore, the validity of using the Sudoscan in evaluating autonomic neuropathy in patients with diabetes has been proven in several studies [3,4]. Since it is well known that autonomic impairment in PD occurs primarily due to ganglionic and postganglionic lesions [8], the Sudoscan should be useful in estimating autonomic function in patients with PD. Work by Xu [23] and our previous work [24] have proved this notion to be correct.

Surprisingly, the number of patients with AN largely increased from 22 to 40 when the Sudoscan-based sudomotor subscore was added, meaning that 18 patients who were initially considered as non-AN participants did in fact have AN. Some of these patients had cardiovagal and adrenergic subscores of 0, but had sudomotor subscores of 2 or more. Thus, they were initially considered as having “normal” autonomic function, and subsequently, were diagnosed with AN. Thus, the prevalence of AN could be underestimated when we focus only on cardiovascular function, i.e., the cardiovagal and adrenergic domains. In addition, the ESC values measured using the Sudoscan were similar between groups when separating patients only by their CASS subscores. These findings suggest that the decline in autonomic function is not parallel between the different domains. The correlation between the COMPASS 31 and CASS scores was not high according to the results of the current study. In contrast to our result, the study by Kim et al. showed that the total COMPASS 31 score was well correlated with the results of the objective autonomic function test [7]. That study did the COMPASS 31 on patients with parkinsonism, including PD and multiple system atrophy. The low correlation in our study could be due to the narrow spectrum of the severity of our enrolled patients. Most of the patients had mild to moderate severity, both in functional status and in autonomic function. Nevertheless, the correlation between the orthostatic intolerance score and the modified CASS score and CASS subscores was statistically significant. In a study by Suarez et al., it was also shown that the orthostatic intolerance score was the best for predicting the CASS score, although they used the original version of COMPASS rather than the abbreviated COMPASS 31 version [5]. By adding the subscore in the sudomotor domain based on the Sudoscan, the correlation between the CASS and the parameters of autonomic function remained preserved; moreover, the correlation coefficient between the CASS and the COMPASS 31 total weighted score increased and the *p*-value decreased (from 0.019 to 0.007). This means that combining the scores in the three domains better reflects the “whole picture” of autonomic function rather than using only those in the cardiovagal and the adrenergic domains.

## 5. Study Limitations

Our study had some limitations. The study only enrolled patients with mild-to-moderate severity. Patients with a broader spectrum of severity should have been recruited to prove that the scoring system could be used in patients with different severity levels. In addition, the sample size was somewhat small considering the prevalence of PD. Furthermore, if we had performed the QSART for these patients to compare the modified CASS with the original CASS, the results would be more convincing. Unfortunately, the QSART in our laboratory was out of order for the above-mentioned reasons. Further, the governments in several regions, such as Taiwan, provided 99% of the nation’s population with risk-sharing insurance for comprehensive medical coverage, including for disease prevention and patient care. However, they do not provide funds for the use of the QSART to screen and diagnose AN; thus, it is not easily available at hospitals in Taiwan.

## 6. Conclusions

Our study demonstrated that the modified CASS can not only better reflect the exact autonomic function, but also improve the characterization and quantification of AN in patients with PD. In areas in which the QSART facilities are not available, the Sudoscan could be a suitable substitution and a time-saving procedure, and could ultimately provide a more comprehensive autonomic testing battery.

## Figures and Tables

**Figure 1 jcm-12-01517-f001:**
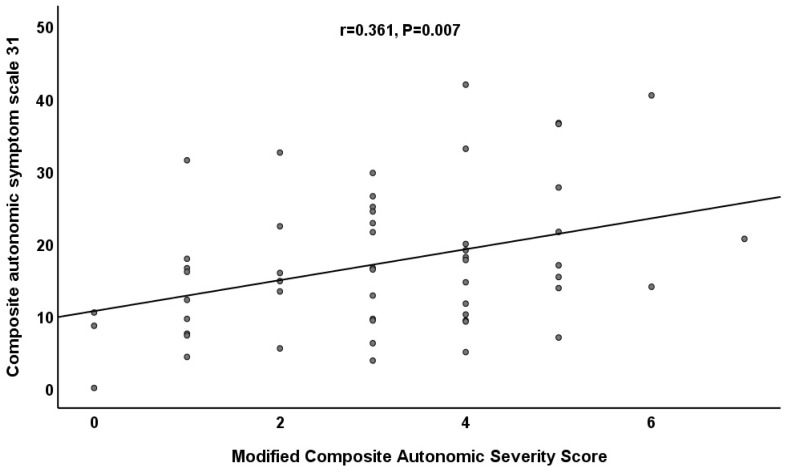
The correlation between the total weighted score of COMPASS 31 and the modified CASS.

**Figure 2 jcm-12-01517-f002:**
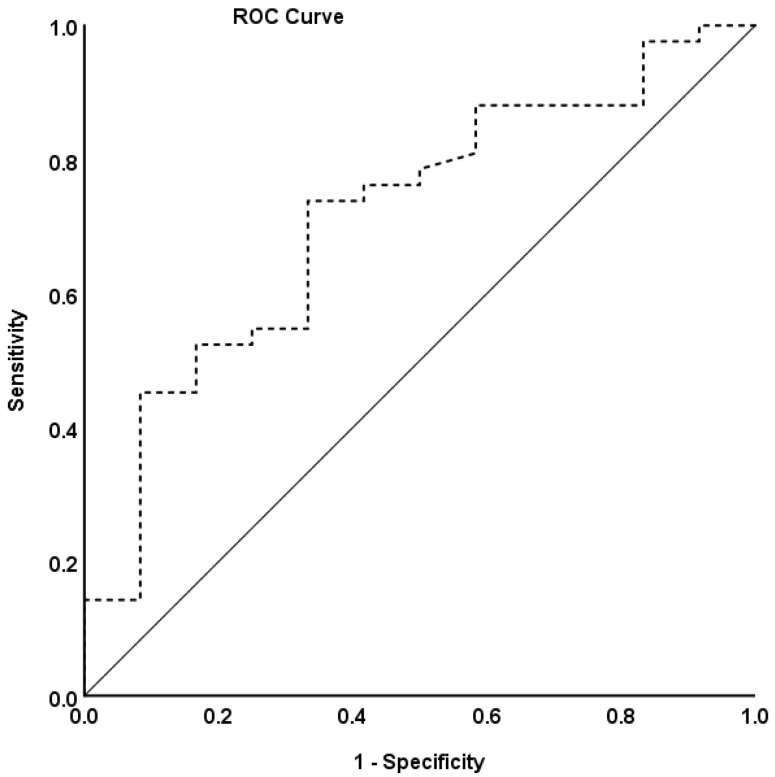
Receiver operator characteristic curves for predicting cardiovascular autonomic neuropathy. AN diagnostic accuracy is shown based on the receiver operating characteristic curve analysis.

**Table 1 jcm-12-01517-t001:** Comparison of original CASS, modified CASS, and CASS subscores.

	Original CASS	Modified CASS	CASS Subscores
**Sudomotor subscore**	QSART	Sudoscan	Omitted
1	Single site abnormal and ≥50% of the lower limit	Only one abnormal mean value of the hand ESC or foot ESC; none of them <50% of the lower limit	
2	Single site <50% of the lower limit, or two or more sites reduced and ≥50% of the lower limit	Both mean values of hand ESC and foot ESC are abnormal but >50% of the lower limit, or one abnormal mean value of hand ESC or foot ESC <50% of the lower limit with the other within the normal limit	
3	Two or more sites <50% of the lower limit	Both mean values of hand ESC and foot ESC are abnormal, and at least one abnormal mean value <50% of the lower limit	
**Cardiovagal subscore**			
0	Normal	The same as the original CASS	The same as the original CASS
1	HRDB mildly reduced but >50% of the minimum		
2	HRDB reduced to <50% of minimum or HRDB + VR reduced		
3	Both HRDB and VR reduced to <50% of the minimum		
**Adrenergic subscore**			
0	Normal	The same as the original CASS	The same as the original CASS
1	Early phase II reduction >20 but <40 mmHg MBP (30–40 if >50 years) Late phase II does not return to the baseline Pulse pressure reduction to ≤50% of baseline		
2	Early phase II reduction >40 mmHg MBP		
3	Early phase II reduction >40 mmHg + absent late phase II and phase IV		
4	Criteria for 3 + orthostatic hypotension (SBP decrease ≥30 mmHg or MBP ≥ 0 mmHg)		

Abbreviations: CASS = Composite Autonomic Scoring Scale; QSART = quantitative sudomotor axonal reflex test; ESC = electrochemical skin conductance; HRDB = heart rate response to deep breathing; VR = Valsalva ratio; SBP = systolic blood pressure; MBP = mean blood pressure.

**Table 2 jcm-12-01517-t002:** Baseline characteristics of Parkinson’s disease.

	Number of Cases (n = 55)
Age (years)	65.5 ± 8.3
Sex (female/male)	26/29
Body mass index (kg/m^2^)	24.8 ± 4.4
Waist circumference (cm)	87.4 ± 9.9
Disease duration (years)	4.8 (2.1, 9.6)
Levodopa equivalent dose (mg/day)	713 (375, 1304)
UPDRS total score	27 (18, 36)
UPDRS I (mentation, behavior, and mood)	1 (0, 2)
UPDRS II (activities of daily living score)	11 (5, 14)
UPDRS III (motor score)	16 (9, 21)
Composite autonomic symptom scale 31	
Orthostatic intolerance	0 (0, 2)
Vasomotor score	0 (0, 0)
Secretomotor score	2 (1, 3)
Gastrointestinal symptoms score	5 (2, 9)
Bladder score	2 (0, 2)
Pupillomotor score	5 (3, 6)
Total weighted COMPASS score	15.9 (9.5, 22.3)
Anti-Parkinsonian medications ^Φ^	
Levodopa	51
Dopamine agonist (pramipexole/ropinirole)	39
MAO-B inhibitors (selegiline/rasagiline)	19
COMT inhibitors (entacapone)	10
Amantadine	6

^Φ^ = All the patients took more than one kind of anti-Parkinsonian medication. Abbreviations: UPDRS **=** Unified Parkinson’s Disease Rating Scale; LED = levodopa equivalent dose, area (cm^2^); MAO-B = monoamine oxidase B; COMT = catechol-o-methyl-transferase.

**Table 3 jcm-12-01517-t003:** Correlation analysis between parameters of COMPASS-31 and CASS subscores and Modified CASS.

Spearman Correlation	CASS Subscores		Modified CASS	
	r	*p*	r	*p*
Composite autonomic symptom scale 31				
Orthostatic intolerance	0.365	0.006 *	0.287	0.035 *
Vasomotor score	0.062	0.654	0.238	0.084
Secretomotor score	0.049	0.723	0.203	0.141
Gastrointestinal symptoms score	0.040	0.769	0.135	0.329
Bladder score	0.273	0.044 *	0.319	0.019 *
Pupillomotor score	0.004	0.979	−0.070	0.616
Total weighted COMPASS 31 score	0.316	0.019 *	0.361	0.007 *

* *p*-value < 0.05. Abbreviations: subscore CASS = cardiovagal and adrenergic subscores; COMPASS 31 = Composite autonomic symptom scale 31.

**Table 4 jcm-12-01517-t004:** Comparison of baseline characteristics and parameters of autonomic function between AN and non-AN groups according to CASS subscores and modified CASS scoring system.

	CASS Subscores Scoring System			Modified CASS Scoring System		
	AN (n = 22)	Non-AN (n = 33)	*p*-Value	AN (n = 40)	Non-AN (n = 15)	*p*-Value
Age (years)	66.3 ± 8.3	64.9 ± 8.4	0.546	66.4 ± 8.2	63.0 ± 8.5	0.193
Sex (female/male)	9/13	17/16	0.444	19/21	7/8	0.476
Body mass index (kg/m^2^)	25.5 ± 4.3	24.3 ± 4.5	0.322	24.7 ± 4.6	25.1 ± 4.0	0.748
Disease duration (years)	5.4 9 (2.2, 9.9)	4.8 (1.8, 9.4)	0.790	4.5 (1.7, 8.8)	5.6 (2.8, 11.9)	0.321
Levodopa equivalent dose (mg)	775 (497, 1337)	613 (328, 1075)	0.189	708 (459, 1307)	880 (150, 1050)	0.461
**Disease severity scale**						
Hoehn and Yahr stage	1.5 (1.4, 3.0)	1.5 (1.3, 2.5)	0.627	1.5 (1.5, 2.9)	1.5 (1.0, 2.5)	0.831
UPDRS I	1.5 (1. 3)	1 (0, 2)	0.179	2 (1, 3)	1 (0, 2)	0.062
UPDRS II (ADL score)	11 (6.8, 14)	8 (5, 15.5)	0.575	11 (6, 13.7)	7 (3, 18)	0.747
UPDRS III (motor score)	16 (13.5, 20.5)	15 (8.5, 21)	0.353	15.5 (12, 19.8)	16 (8, 26)	0.842
**COMPASS 31**						
Orthostatic intolerance	0 (0, 5)	0 (0, 0)	0.004 *	0 (0, 3)	0 (0, 0)	0.031 *
Vasomotor score	0 (0, 0)	0 (0, 0)	0.791	0 (0, 0)	0 (0, 0)	0.382
Secretomotor score	2 (0.8, 3)	2 (1.5, 3)	0.986	3 (2, 3)	2 (0, 2)	0.111
Gastrointestinal symptoms score	4 (2.8, 9)	5 (1, 8.5)	0.938	6 (3, 9)	5 (0, 7)	0.235
Bladder score	2 (1, 3)	1 (0, 2)	0.082	2 (1, 2)	0 (0, 2)	0.054
Pupillomotor score	5 (3, 6)	5 (3, 6)	0.566	5 (3, 6)	5 (3, 6)	0.497
Total weighted COMPASS score	19.0 (12.5, 29.0)	14.6 (8.9. 18.4)	0.026 *	17.3 (11.9, 24.8)	9.5 (6.2. 16.0)	0.003 *
**Autonomic function**						
HR_DB (beats/min)	5.6 ± 1.7	10.4 ± 5.1	<0.0001 *	7.1 ± 3.1	11.5 ± 6.4	0.022 *
Valsalva ratio	1.24 ± 0.14	1.39 ± 0.16	0.001 *	1.30 ± 0.15	1.39 ± 0.20	0.131
BRS_VM	1.3 ± 0.9	2.3 ± 1.2	0.003 *	1.7 ± 1.1	2.2 ± 1.2	0.206
Mean hand ESC (µS)	47.1 ± 18.6	44.1 ± 19.7	0.565	40.5 ± 18.0	58.1 ± 16.3	0.002 *
Mean foot ESC (µS)	52.0 ± 17.8	53.6 ± 17.8	0.729	47.0 ± 16.3	68.8 ± 9.8	0.000 *
ESC CAN risk score	34.1 ± 5.2	31.9 ± 6.6	0.173	33.9 ± 5.4	29.9 ± 6.9	0.028 *

Values are expressed as mean ± SD or median (interquartile range (IQR)). * = It indicates *p* < 0.05. Abbreviations: UPDRS = Unified Parkinson’s Disease Rating Scale; ADL = activities of daily living; COMPASS 31 = composite autonomic symptom scale 31; HR_DB = heart rate response to deep breathing; BRS_VM = baroreflex sensitivity obtained via Valsalva maneuver; ESC = electrochemical skin conductance; CAN = cardiovascular autonomic neuropathy.

**Table 5 jcm-12-01517-t005:** Sensitivity, specificity, and area under the curve obtained using ROC analysis for the use of composite autonomic symptom scale 31 in predicting modified CASS-based cardiovascular autonomic neuropathy.

Variables	Cut-Off Value *	AUC (95% CI)	Sensitivity (%)	Specificity (%)	*p*-Value
COMPASS 31	12.45	0.715 (0.55–0.88)	74	67	0.02 *

* = *p* < 0.05; Abbreviations: ROC = receiver operating characteristic; COMPASS 31 = composite autonomic symptom scale 31.

## Data Availability

The datasets used and/or analyzed during the current study are available from the corresponding author upon reasonable request.

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
