# Peer review of "Assessing the Feasibility of Using Electrochemical Skin Conductance as a Substitute for the Quantitative Sudomotor Axon Reflex Test in the Composite Autonomic Scoring Scale and Its Correlation with Composite Autonomic Symptom Scale 31 in Parkinson’s Disease"

_jcm, 2023, doi:10.3390/jcm12041517_

Round 1

Reviewer 1 Report

With respect to the manuscript entitled: “Feasibility of electrochemical skin conductance substitute 2 quantitative sudomotor axon reflex test in composite auto-3 nomic scoring scale and assessing its correlation with compo-4 site autonomic symptom scale 31 in Parkinson's disease”, I think that it is an interesting subject and important for Health professionals as the results can contribute to improve the characterization and quantification of CAN in patients with PD.

However, I think that clarification about some aspects of the trial is needed:

1-      The writing of the manuscript should be improved. It is important to make it more comprehensible

2-      The Introduction and discussion should be supported with more references to make the importance of the study as well as the benefits to use the modified scale more sustainable.

3-      The limitations of the study have to be taken into account in the conclusions

Author Response

Reviewer 1)

Comments and Suggestions for Authors

With respect to the manuscript entitled: “Feasibility of electrochemical skin conductance substitute 2 quantitative sudomotor axon reflex test in composite auto-3 nomic scoring scale and assessing its correlation with compo-4 site autonomic symptom scale 31 in Parkinson's disease”, I think that it is an interesting subject and important for Health professionals as the results can contribute to improve the characterization and quantification of CAN in patients with PD. However, I think that clarification about some aspects of the trial is needed:

  • The writing of the manuscript should be improved. It is important to make it more comprehensible.

Answers: Thanks for your suggestion. We will have our manuscript undergo English editing service provided by MDPI to make it more comprehensible.

  • The Introduction and discussion should be supported with more references to make the importance of the study as well as the benefits to use the modified scale more sustainable.

Answers: Some discussion with references was added to the first paragraph (P.8, L.202-208) of DISCUSSION section to enhance the benefit of using modified CASS.

  • The limitations of the study have to be taken into account in the conclusions

Answers: Thanks for your comment. We revised the sentence in the conclusion section to your comment. They are as follows:

In those areas in which the facility of QSART is not available, Sudoscan could be a substitution and a time-saving procedure, but ultimately more time-consuming, complete autonomic function testing.

Reviewer 2 Report

Reviewers' comments to Authors:

The study addresses an important issue that investigated feasibility of the 19 electrochemical skin conductance (Sudoscan) substitute quantitative sudomotor axon reflex test (QSART) as the sudomotor domain and assessed its correlation with COMPASS 31 in patients with 21 Parkinson’s disease (PD)

This is very interesting study, however authors should consider the following comments:

-       Introduction is well written however, authors should add more information about current state of knowledge  of using CASS and COMPASS-31 in Parkinson’s studies.

-       There  is no aim of the study at the end of introduction

-       Autonomic function tests  and Compass-31 scale should be shortly described

-       Authors didn’t mention about BRS, these information firstly appear in results paragraph

-       In Discussion section authors should include other results from electrophysiological studies  comparing both methods (Compass-31, CASS)

-       The sample size is relative in terms of prevalence of PD, this is limitation of the study

Author Response

Comments and Suggestions for Authors

The study addresses an important issue that investigated feasibility of the 19 electrochemical skin conductance (Sudoscan) substitute quantitative sudomotor axon reflex test (QSART) as the sudomotor domain and assessed its correlation with COMPASS 31 in patients with 21 Parkinson’s disease (PD)

This is very interesting study; however, authors should consider the following comments:

  • Introduction is well written however, authors should add more information about current state of knowledge of using CASS and COMPASS-31 in Parkinson’s studies.

Answers: In the section of introduction, several references have been added to demonstrate the usefulness of CASS and COMPASS 31 in patients with parkinsonism (P.2, L65-67; The references 7 and 8 were newly added.).

  • There is no aim of the study at the end of introduction

Answers: We have added the aim of the study in the last sentence of the INTRODUCTION section (P.2, L.74-76).

  • Autonomic function tests and Compass-31 scale should be shortly described

Answers: The autonomic test battery consisted deep breathing, Valsalva maneuver, and 5-minute head-up tilt tests. It has been prescribed in section 2.3 (P.3). The introduction to COMPASS 31 was added in INTODUCTION section. (P.2, L.60-65)

  • Authors didn’t mention about BRS, this information firstly appears in results paragraph

Answers: The following statement concerning BRS was added to section 2.3 on P.3 (L.114-116):

“Baroreflex sensitivity (BRS) was derived from changes in HR and blood pressure during early phase II of VM by applying least-squares regression analysis (BRS_VM).”

  • In Discussion section authors should include other results from electrophysiological studies comparing both methods (Compass-31, CASS)

Answers: We added a paper by Kim et. al. as a reference. In contrast to our result, their study showed that the total COMPASS 31 score was well correlated with the results of objective autonomic function test. Then we discussed the discrepancy between these two study results. (P.8, L.241-244)

  • The sample size is relative in terms of prevalence of PD, this is limitation of the study

Answers: In Discussion section, we have added the small sample size as the limitation of the study (P.8, L.261).

Reviewer 3 Report

Very good paper, interesting research well done, data analisis and discusion are clear and significant

Author Response

Reviewer 3)

Very good paper, interesting research well done, data analysis and discussion are clear and significant

Answers: Thanks for your comment.
